# The timing of transcription of RpoS-dependent genes varies across multiple stresses in *Escherichia coli* K-12

Josephine Adams,[1] Johnson Hoang,[1] Emily Petroni,[1] Ethan Ashby,[2] Johanna Hardin,[2] Daniel M. Stoebel[1]

**ABSTRACT** The alternative sigma factor RpoS regulates transcription of over 1,000 genes in *Escherichia coli* in response to many different stresses. RpoS levels rise continuously after exposure to stress, and the consequences of changing levels of RpoS for the temporal patterns of expression of RpoS-regulated genes have not been described. We measured RpoS levels at various times during the entry to stationary phase, or in response to high osmolarity or low temperature, and found that the time required to reach maximum levels varied by several hours. We quantified the transcriptome across these stresses using RNA-seq. The number of differentially expressed genes differed among stresses, with 1,379 DE genes identified in stationary phase, 633 in high osmolarity, and 302 in cold shock. To quantify the timing of gene expression, we fit sigmoid or double sigmoid models to differentially expressed genes in each stress. During the entry into stationary phase, genes whose expression rose earlier tended to be those that had been found to respond most strongly to low levels of RpoS. The timing of individual gene's expression was not correlated across stresses. Taken together, our results demonstrate *E. coli* activates RpoS with different timing in response to different stresses, which in turn generates a unique pattern of timing of the transcription response to each stress.

**IMPORTANCE** Bacteria adapt to changing environments by altering the transcription of their genes. Specific proteins can regulate these changes. This study explored how a single protein called RpoS controls how many genes change expression during adaptation to three stresses. We found that: (i) RpoS is responsible for activating different genes in different stresses; (ii) that during a stress, the timing of gene activation depends on the what stress it is; and (iii) that how much RpoS a gene needs in order to be activated can predict when that gene will be activated during the stress of stationary phase.

**KEYWORDS** stress response, transcription, RpoS, RNA-seq

Many bacteria live in changing environments and face fluctuating stresses. A primary way they respond to stresses is by changing the pattern of transcription of their genes (1). In *Escherichia coli* and related enteric bacteria, the alternative sigma factor RpoS is a key global regulator of this stress response. This general stress response is induced by many different physical stresses: stationary phase; starvation for carbon, phosphate, and magnesium; elevated osmolarity; low temperature; and DNA damage (reviewed in references 1, 2). In response to these and other stressors, RpoS protein accumulates in cells.

RpoS concentration in the cell is regulated at many levels (reviewed in references 1, 2). Transcription of the *rpoS* gene is directly regulated by the transcription factor ArcA (3). Translation of the rpoS mRNA is regulated by the small RNAs dsrA, rprA, and arcZ together with the protein Hfq (4–6). RpoS is degraded by the ClpXP protease (7).

Address correspondence to Daniel M. Stoebel, stoebel@g.hmc.edu.

Josephine Adams, Johnson Hoang, and Emily Petroni contributed equally to this article. They are ordered alphabetically by last name.

The authors declare no conflict of interest.

See the funding table on p. 14.

Proteolysis requires the RssB adaptor protein (8–10), and proteolysis can be blocked by antiadaptor proteins IraD, IraM, and IraP (11, 12). The activity of the RpoS protein can also be regulated by other molecules like the protein Crl (13). The plethora of regulatory inputs allows RpoS to be activated to different extents in response to different stresses (14).

RpoS regulates the transcription of a large number of genes by recruiting RNA polymerase to specific promoters. In stationary phase in *E. coli*, 23% of all genes in the genome are directly or indirectly regulated by RpoS (15). While many different stresses lead to the accumulation of RpoS in the cell, the transcriptional response differs across these stresses. For example, Weber et al. (16) found that of 481 genes differentially expressed across the entry into stationary phase, osmotic upshift, and acid stress, only 140 (29%) were differentially expressed in all three conditions. The existence of different RpoS regulons in response to different stresses suggests that other components of the RpoS response could differ as well.

RpoS levels rise continuously following onset of stress. It is unknown how the levels of RpoS affect the timing of expression of genes in its regulon. There are well-described regulatory systems where promoters that can be strongly transcribed with low levels of a regulator are transcribed before those that require higher levels of that regulator. This is the case for Spo0A, a master regulator of sporulation in *Bacillus subtilis* (17). Genes that only require low levels of Spo0A to be transcribed are involved in physiological responses that occur early in response to starvation. Genes that require high levels of Spo0A are transcribed after more prolonged starvation, when cells begin a sporulation program. PhoB's role in the response to phosphate starvation in *E. coli* shows similar patterns to Spo0A in *B. subtilis* (18). In this system, the genes that are expressed first in response to phosphate starvation are genes with promoters that have high affinity for PhoB, and hence require lower amounts of PhoB for expression, compared to those expressed later that have lower affinity for PhoB and so require higher levels of PhoB in the cell for expression. Similar to the Spo0A case, there are distinct physiological differences between early and late PhoB-regulated genes.

To study the relationship between RpoS levels and transcription, in earlier work we controlled the level of expression of RpoS in stationary phase and measured the transcriptional response using RNA-seq (15). We classified genes that achieved high levels of transcription in response to low amounts of RpoS as sensitive, and those genes that required high levels of RpoS for high levels of transcription as insensitive to RpoS. Using a comparison to an existing RNA-seq time-course data set (19), we found that sensitive genes were transcribed earlier in the transition to stationary phase than insensitive genes were (15). However, the low number of timepoints and lack of replication of individual timepoints in the earlier data set made it impossible to quantify the magnitude of any difference in timing of transcription. In addition, it was unclear whether the tentative relationship found in the stress response to stationary phase would hold across other stresses that also activate RpoS.

In this study, we hypothesized that genes from different RpoS sensitivity classes have different average times that they are expressed during the transition to stationary phase, and that the difference in timing is large enough to be physiologically meaningful. In addition, we wished to explore if there was a similar timing difference during the response to other stresses. To address these questions, we used RNA-seq to measure the levels of expression of the RpoS regulon during three different stresses: the transition to stationary phase, in response to high osmolarity, and to low temperature.

## MATERIALS AND METHODS

### Strains and growth conditions

The wild-type and Δ*rpoS E. coli* strains used for these experiments (Table 1) have been used for previous experiments in our laboratory (15).

**TABLE 1**  Strains used in RNA-seq and protein isolation experiments

| Published designation | Genotype | Reference |
|---|---|---|
| DMS2537 | F-, Δ(araD-araB)567, ΔlacZ4787(::rrnB-3), λ-, Δ(araH-araF)570(::FRT), ΔaraEp-532::FRT, φPcp13araE534, Δ(rhaD-rhaB)568, hsdR514 | (15) |
| DMS2545 | ΔrpoS mutant (F-, Δ(araD-araB)567, ΔlacZ4787(::rrnB-3), λ-, Δ(araH-araF)570(::FRT), ΔaraEp-532::FRT, φPcp13araE534, Δ(rhaD-rhaB)568, hsdR514, ΔrpoS746::kan) | (15) |

For protein and RNA isolation, DMS2537 and DMS2545 were inoculated from −80°C frozen cultures and grown aerobically overnight in 5 mL of Luria–Bertani (LB) broth (1% tryptone, 0.5% yeast extract, 1% NaCl) at 37°C with shaking at 225 rpm. The next morning, 250 µL of each culture was diluted in 25 mL of LB and grown at 37°C in a shaking water bath at 225 rpm until the $OD_{600}$ of the culture reached approximately 0.3 ($OD_{600}$ of 0.3 is in the middle of exponential growth in these cultures). At this point, samples were taken for RNA and protein isolation and then the culture was exposed to the stressor of interest: for low temperature, they were moved to a 15°C shaking water bath; for high osmolarity, 3 mL of 5 M NaCl was added to the cultures; or they were left to continue growing to reach stationary phase. This method of osmotic stress results in a small difference in total culture volume and nutrient concentration relative to the other two conditions.

## Measurements of growth parameters

In order to estimate growth parameters during all three stresses, $OD_{600}$ was measured every 15 min for 150 min for the high-osmolarity condition, every 15 min for 300 min for the entry into stationary phase condition, and every 30 min for 300 min for the low temperature condition. These times were chosen to match the time spans chosen for protein and RNA measurements (below). $N = 5$ replicates were performed for each strain in each condition.

Growth parameters were estimated using the package growthrates in R (20). A logistic model was used for the entry to stationary phase data, as these cultures were already actively growing at the start of data collection and there was no lag phase. For the high osmolarity and low-temperature cultures, the Gomperz three parameter model was fit in order to estimate both lag time and exponential growth rate. These cultures did not reach stationary phase in the time period of interest so the density at stationary phase is not reported. Estimates of mumax were converted to doubling times by dividing ln (2) by mumax.

## Culture growth for RNA-seq and western blot samples

Two sets of growth experiments were performed to collect samples for RNA-seq and western blotting experiments. First, $OD_{600}$ measurements, protein, and RNA samples were taken every half hour for 150 min for all three stressors ($n = 3$ replicates for each strain, with each replicate on a different day). Additional longer experiments were also performed for stationary phase and low temperature in order to capture more of the change in RpoS levels in response to the stress. To gather data for the entry to stationary phase, another set of measurements and samples were taken every 60 min for 300 min ($n = 3$ replicates per strain, with each replicate on a different day). To gather data for low temperature, another set of samples were taken at 90, 120, 180, 240, and 300 min ($n = 4$ replicates per strain, with each replicate on a different day). No longer experiments were performed for the high-osmolarity condition, because during longer time periods cultures would begin entering stationary phase. We wished to study osmotic-induced RpoS expression during exponential growth, rather than the combination of osmotic and stationary phase induction of RpoS. In total, there were three sets of samples for the high

osmolarity treatment, six sets of samples for the entry to stationary phase, and seven sets of samples for low temperature.

## Protein isolation and western blotting

Samples with cell density equivalent to 1 mL of $OD_{600}$ = 0.3 were pelleted at, at least, 20,000 × g for 30 s, resuspended in 100 µL of 2× Laemmli sample buffer (LiCor), heated at 95°C for 5 min, and then stored at −20°C until they were used.

A quantitative western blot was used to measure the levels of RpoS in the cell at each time point, in a procedure slightly modified from reference 15. Ten microliters of each protein sample were electrophoresed on a 4–20% gradient polyacrylamide gel (Bio-Rad) in Tris-glycine running buffer (25 mM Tris base, 250 mM glycine, 0.5% SDS) at 200V for 35 min at room temperature. Electrophoresis was used to transfer proteins to a Immobilon-FL PVDF membrane in transfer buffer (48 mM Tris base, 39 mM glycine, 0.0375% SDS, 20% methanol) at 100 V for 45 min at 4°C. In order to quantify the total amount of protein in the samples, the membranes were stained with 5 mL of REVERT Total Protein Stain solution and imaged using the LiCor Odyssey CLx imager. The membranes were blocked with 10 mL of Odyssey Blocking Buffer for 1 h at room temperature.

RpoS was detected using an anti-RpoS monoclonal antibody (clone 1RS1, BioLegend). The blocked membranes were labeled with primary antibody (0.4 µg/mL mouse anti-RpoS, 10 mL Odyssey Blocking Buffer, 0.2% Tween 20) at 4°C with shaking at 55 rpm overnight. Once labeled, the membrane was washed four times with 10–15 mL of 1× Tris-buffered saline with Tween 20 for 5 min each wash. The membranes were probed with a fluorescent secondary antibody (IRDye 800CW goat anti-mouse, LiCor) that was diluted in 10 mL of Odyssey Blocking Buffer with 0.2% Tween 20 and 0.01% SDS. The membranes were incubated with the diluted secondary antibody for 1 h at room temperature and were washed as described previously, with an additional wash at the end with 1× Tris-buffered saline without Tween 20. Membranes were dried for 2–3 h between two sheets of Whatmann 3 MM blotting paper before imaging on a Li-Cor Odyssey CLx imager.

RpoS levels were quantified by analyzing the amount of fluorescence of each band with Image Studio v2.1 software. Data analysis was performed in R v4.1.2 (21). RpoS levels were normalized by dividing them by the total amount of protein at each time point, which was found in the quantification of the total protein stain. Loess regression was applied to the normalized levels to examine the relationship between RpoS levels and time. The loess model provided the maximal RpoS levels and the first time corresponding to half the maximal level.

## RNA isolation

Two hundred microliters of culture were pelleted and resuspended in 500 µL of Trizol preheated to 65°C and then stored at −80°C until use. Samples were purified on a column using the Zymo Direct-Zol RNA Miniprep kit. They were then treated twice with DNase I using Turbo DNA-free (Invitrogen) for 30 min each at 37°C to ensure that all DNA was degraded. The samples were further purified on a column using the Zymo RNA Clean and Concentrator Kit and again stored at −80°C until sequencing. The concentration of the samples was checked using a NanoDrop spectrophotometer and degradation was checked for by electrophoresing samples on an ethidium bromide agarose gel.

## Generation of RNA-seq data

Illumina cDNA libraries were generated using a modified version of the RNAtag-seq protocol (22). Briefly, 500 ng to 1 µg of total RNA was fragmented by heating to 94°C in NEB Alkaline Phosphatase Buffer, depleted of any residual genomic DNA, dephosphory-lated, and ligated to DNA adapters carrying 5′-$AN_8$-3′ barcodes of known sequence with a 5′ phosphate and a 3′ blocking group. Barcoded RNAs were pooled and depleted of rRNA using the RiboZero rRNA depletion kit (Epicentre). Pools of barcoded RNAs

were converted to Illumina cDNA libraries in two main steps: (i) reverse transcription of the RNA using a primer designed to the constant region of the barcoded adaptor with addition of an adapter to the 3′ end of the cDNA by template switching using SMARTScribe (Clontech) as described (23) and (ii) PCR amplification using primers whose 5′ ends target the constant regions of the 3′ or 5′ adaptors and whose 3′ ends contain the full Illumina P5 or P7 sequences. cDNA libraries were sequenced on the Illumina (NextSeq 500) platform at the Broad Institute to generate paired end reads.

### RNA-seq data analysis

Sequencing reads from each sample in a pool were demultiplexed based on their associated barcode sequence using custom scripts. Up to one mismatch in the barcode was allowed provided it did not make assignment of the read to a different barcode possible. Barcode sequences were removed from the first read as were terminal Gs from the second read that may have been added by SMARTScribe during template switching.

Reads were aligned to the NCBI *E. coli* K-12 MG1655 genome (RefSeq assembly GCF_000005855.2) using BWA v0.7.17 (24). Reads were counted with htseq-count v2.0.1 (25). Further analysis was done in R v4.1.2 (21). Differential expression analysis was performed using the package DESeq2 v1.34.0 (26). Testing for genes that changed with time in a RpoS-dependent manner was done with a likelihood ratio test, comparing a full model with genotype, time, and genotype × time interaction with a reduced model with only genotype and time effects. The false discovery rate (FDR) was controlled by adjusting *P*-values using the method of Benjamini–Hochberg (27).

RNA-seq time-course data were modeled with the R package Sicegar v0.2.4 (28). All analyses used the default parameters, except that the threshold intensity ratio was set to 0.65 and the threshold AIC was set to 5. The stationary phase and cold data sets combined two different experiments and sequencing batches. Batch effects were removed on the variance stabilized transformed data using the removeBatchEffects() from the package limma v3.50.3 (29). Data were reverse transformed back to the original scale by applying an exponential function with base 2. The time at which half-maximal RNA abundance was reached (the "onset time") was then extracted from the model for each gene in each stress.

The package infer v1.0.0 was used for permutation-based statistical tests (30).

GO analysis was done using the PANTHER Classification System website (31). A list of genes was uploaded and compared to the entire *E.coli* genome. A statistical overrepresentation test was run using the GO biological process complete annotation set.

## RESULTS

### RpoS expression across stresses

To understand how patterns of RpoS-dependent transcription change in response to the onset of a stress, we chose three stressors that had been previously characterized as inducing RpoS: entry to stationary phase, high osmolarity, and low temperature (32, 33). Starting with *E. coli* K-12 in exponential phase in rich media at 37°C, we induced stress by allowing cells to grow into stationary phase, by raising the NaCl concentration by 600 mM, or by shifting the culture to 15°C. Wild-type and Δ*rpoS* strains grew similarly in all three stresses (Table 2; Fig. S1). For every growth parameter, the mean difference between the two strains was not significantly different (FDR-adjusted *P* > 0.28, permutation test on difference in means, 10,000 permutations, all *P*-values FDR adjusted).

For all three stresses, RpoS levels changed gradually with time (Fig. 1; Fig. S2), but expression kinetics varied across stressors. During the entry to stationary phase, RpoS levels increased for nearly the entire experiment, peaking at 265 min with a normalized RpoS level of 0.047. In the presence of high osmolarity, RpoS levels rose rapidly and peaked at 50 min with a normalized RpoS level of 0.050. In low temperature, RpoS levels peaked at 240 min with a maximum normalized RpoS level of 0.024. Overall, the timing of maximal RpoS levels varies more than the amount of RpoS at the peak.

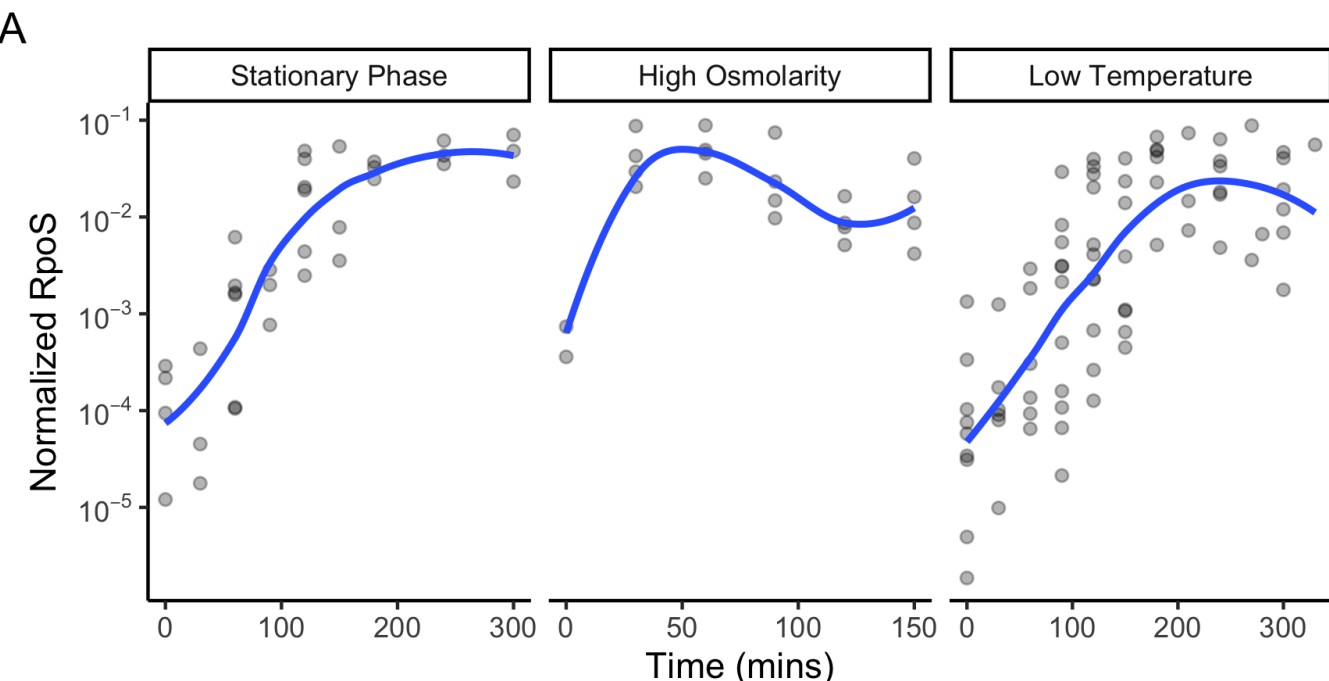

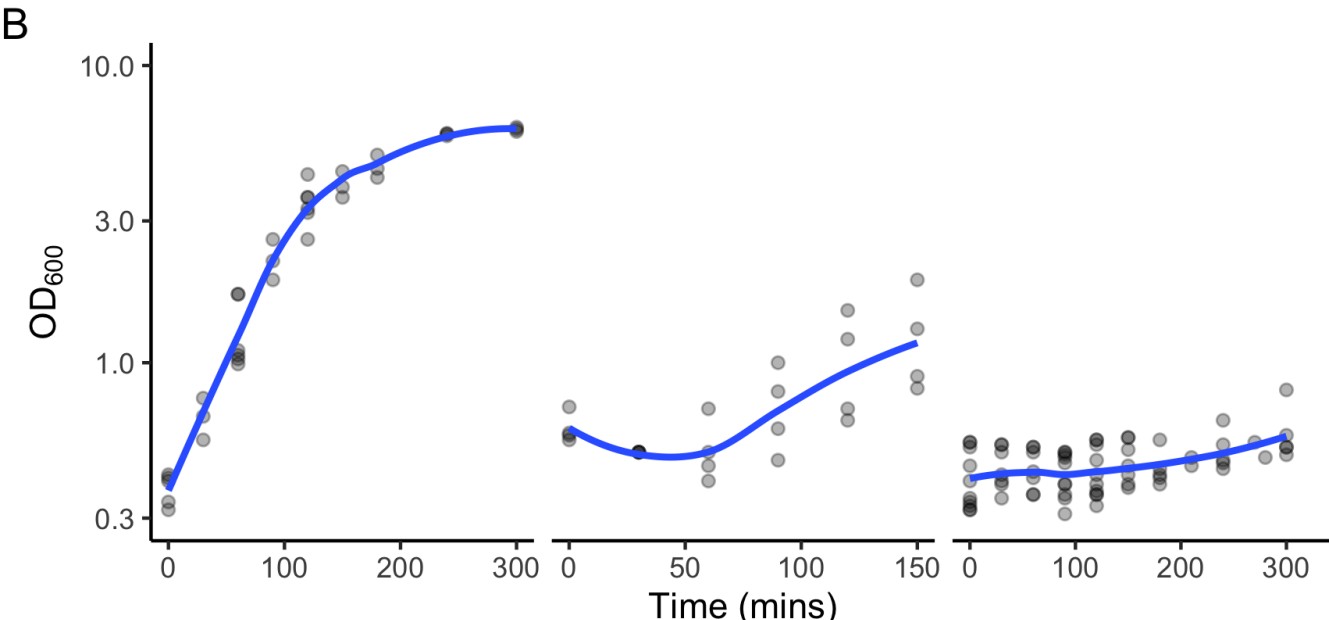

**FIG 1** RpoS levels in *E.coli* in different stressors over time. Levels of (A) RpoS and (B) cell density over time for the three stress conditions in this study, the entry into stationary phase, high osmolarity, and low temperature. RpoS levels were measured by western blot and normalized to total protein level. Cell density was measured by spectrophotometry. Gray dots represent individual measurements and blue curves are loess regressions.

**TABLE 2** Growth parameters of cultures during the three conditions of this study[a]

|  | Entry into stationary phase | | High osmolarity | | Low temperature | |
|---|---|---|---|---|---|---|
|  | Doubling time (min) | Final density ($OD_{600}$) | Doubling time (min) | Lag time (min) | Doubling time (min) | Lag time (min) |
| Wild-type | 30.9 ± 0.5 | 6.2 ± 0.1 | 59.3 ± 0.6 | 33.4 ± 2.1 | 241.7 ± 19.7 | 164.6 ± 23.8 |
| ΔrpoS | 31.1 ± 0.5 | 6.0 ± 0.1 | 53.0 ± 0.9 | 35.6 ± 2.0 | 296.5 ± 13.8 | 127.0 ± 17.2 |

[a]All estimates are mean ± standard error of the mean, *n* = 5 for all values.

## Transcriptome changes during three stresses

To study the role of RpoS in the transcriptional response to the three stresses, we used RNA-seq to measure transcript abundance in wild-type and Δ*rpoS* strains in each condition. In order to study the timing of RpoS-dependent gene expression in these stresses, we first characterized the regulon more generally. We modeled the counts for each gene with a linear model in DESeq2 (26) with terms for time + genotype + time × genotype, and considered as significant those genes with a significant time × genotype interaction term, as judged by a likelihood ratio test (FDR-adjusted *P*-value < 0.01). Less technically, this means that we considered as significant genes where the counts changed across time in a different manner in wild-type compared to Δ*rpoS* strains. The number of differentially expressed (DE) genes differed over 4-fold across the three different stressors, with stationary phase having 1,379 DE genes, high osmolarity 633 DE genes, and low temperature 302 DE genes (Fig. 2; Table S1). The similarity of the two strain's growth parameters (Table 2) implies that differences in growth rate do not cause genes to be DE between wild-type and ΔrpoS strains within a given stress.

One hundred fifty-six genes were DE in all three conditions, but many genes were DE in only a single condition, confirming that there is both a core RpoS-dependent transcriptional response and extensive condition-dependent responses, as first reported by Weber et al. (16). The core 156 genes found in this study include a number of well-studied RpoS-dependent genes, such as *bolA, dps, gadB, gadC, gadX, hdeA, hdeB, hdeD, osmE, osmF, osmY, otsA, otsB, otsA, and rssB* (34–44). Weber et al. (16) found 140 DE genes across stationary phase, high osmolarity, and acidic environment, and 81 genes are shared between our data set and that of Weber et al. (16) In addition, 48 of the 156 genes were identified as directly bound by RpoS in the ChIP-seq analysis of Wong et al. (15). It is not clear if the other genes that are DE in all three conditions are not directly bound by RpoS, or if they were missed in the ChIP-seq experiment.

To better understand the functions of the RpoS-dependent genes under different conditions, we used the PANTHER Classification System to examine which gene functions, as described by the Gene Ontology (GO) vocabulary, are more abundant in certain groups of DE genes. In general, metabolic functions were enriched in all three stresses, although the specific metabolic functions were different. During the entry into stationary phase, 595 genes are annotated as having a cellular metabolic function (Table S2). There are twice as many genes annotated with peptidoglycan metabolic processes in this set of DE genes than the null expectation (FDR-adjusted *P*-value = 0.04) but those genes represent only 41 of the 1,379 DE genes over this transition. In high osmolarity, enriched metabolic functions include those for colanic acid biosynthesis, ATP metabolism, glycolysis, polyol metabolism, and response to heat (FDR-adjusted *P*-value < 0.05; Table S3), all though each of these categories accounted for only 2–3% of all of the DE genes in this condition. In low temperature, genes associated with amino acid catabolism are enriched, but they represent less than 5% of all DE genes in this condition (FDR-adjusted *P*-value = 0.02; Table S4). GO enrichment reveals a few gene functions more abundant than expected in the DE genes but those enriched functions are only associated with a few of the genes.

## The timing of RpoS-dependent gene expression

RpoS protein levels increase at different times in different stresses, and we wanted to explore what this means for the timing of RpoS-dependent gene expression. To quantify the time it takes for a gene to change its expression pattern from "on" to "off" (or vice versa), we attempted to fit sigmoid and double sigmoid models to the expression data using Sicegar (28). A sigmoid curve is S shaped, starting at one level, then changing to another (Fig. 3A). A double sigmoid curve starts at one level, changes in a S-shaped curve to a second level, and then returns back toward the initial level in a second S-shaped curve (Fig. 3B). The Sicegar algorithm tests if the model satisfies criteria related to minimal and maximal expression, expression at the start of the time course, and the model's Akaike Information Criterion (28). If one of the two models (sigmoid or double

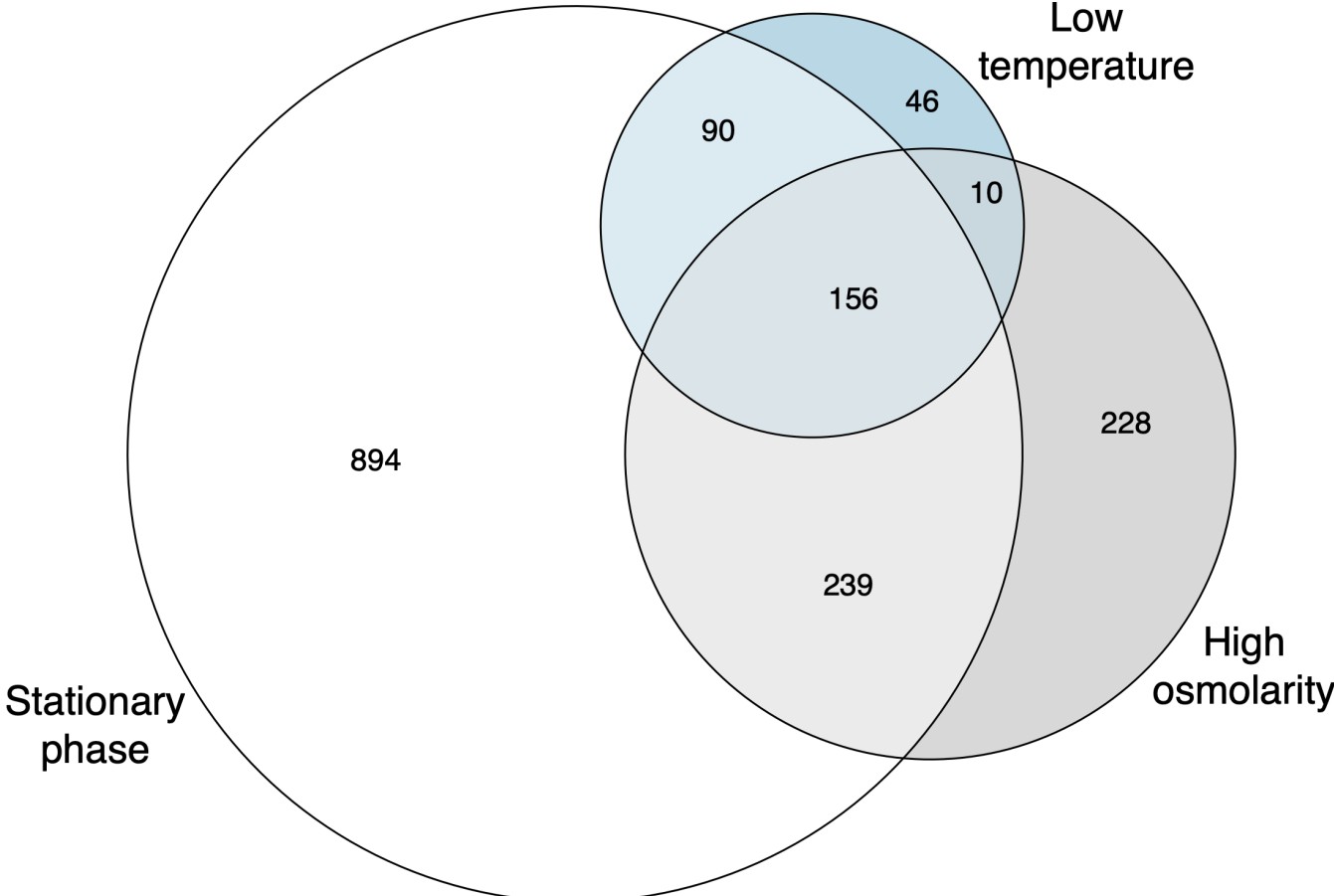

**FIG 2** Venn diagram of DE genes during three different stressors as determined by DESeq2 with an LRT. White represents entry to stationary phase, gray represents high osmolarity, and blue represents low temperature. The overlapping regions indicate how many genes were differentially expressed in the overlapping stressors. In total, 1,379 DE genes were identified in stationary phase, 633 in high osmolarity, and 302 in low temperature.

sigmoid) fit well, we extracted the "onset time," or time at which half maximal transcription is reached. If neither model fit well, the expression pattern for that gene in that condition was labeled as ambiguous (Fig. 3C), and no onset time could be estimated. The data used for the model fitting were the normalized counts of each sample of the wild-type strain for a given condition. We only attempted to fit models for a given gene in a given condition if the gene had earlier been determined to be significantly differentially expressed in the wild-type to Δ*rpoS* comparison.

Sigmoid or double sigmoid curves fit well to about half of the DE genes in each of the conditions. More precisely, 537 of 1,379 (39%) of DE genes during the entry to stationary phase were well-fit by one of the models, as were 278 of 633 (43%) of DE genes during exposure to high osmolarity, and 177 of 302 (59%) of DE genes during exposure to low temperature. Visual inspection of plots of genes for which a sigmoid or double sigmoid model fit poorly showed that most such genes were noisy and generally had low levels of expression. In some cases, the model appeared not to fit well because gene expression was still rising at the last time point, rather than having started to or completely plateaued.

The median onset time of RpoS-regulated genes in the three stresses varied, and all three times are all significantly different from each other (Fig. 4A). In general, the median onset time within each condition was similar to the time at which RpoS reached its half-maximal level (Fig. 4B). The median onset time for all of the genes in stationary phase was 181 min (95% CI: 169–190 min, 1,000 bootstrap replicates), 18 min after RpoS protein levels reached half-maximal levels (163 min). The median onset time for all of the genes

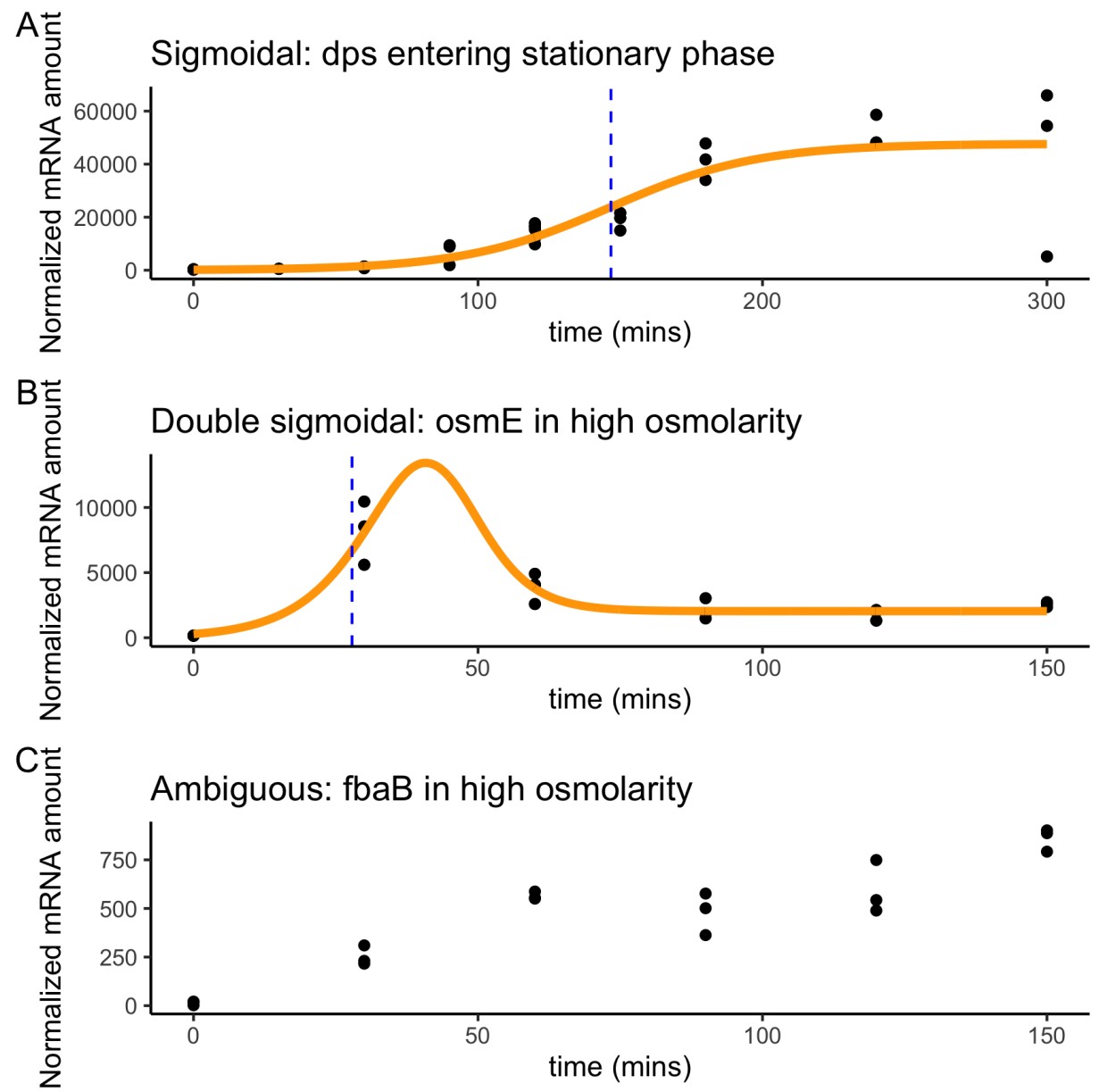

**FIG 3** Examples of the three classifications of individual gene expression patterns. Expression patterns could be classified as either (A) sigmoidal, like *dps* entering stationary phase, (B) double sigmoidal, like *osmE* in high osmolarity, or (C) ambiguous, like *fbaB* in high osmolarity. A model can be fit for the first two cases. Black dots represent individual measurements, the best-fit line is shown in solid yellow, and the onset time (time of half-maximal expression) is shown as the blue dashed line.

in high osmolarity was 31 &&min (95% CI: 30–35 min, 1,000 bootstrap replicates), only 1 min after RpoS protein levels reached their half-maximal levels (at 30 min). The median onset time for all of the genes in low temperature was 140 min (95% CI: 131–149 min, 1,000 bootstrap replicates), which occurred *before* RpoS protein levels reached their half-maximal levels at 172 min. In general, the median onset time of genes is similar to the time that RpoS reached half-maximal levels in a given stress, although there is some difference between environments in the relationship between time to RpoS half-maximal levels and gene onset time.

In earlier work, we categorized RpoS-dependent genes as sensitive, linear, or insensitive based on how they responded to various levels of RpoS in stationary phase (15). We predicted that sensitive genes should turn on before linear genes, which should turn on before insensitive genes. This prediction was correct in stationary phase (Fig.

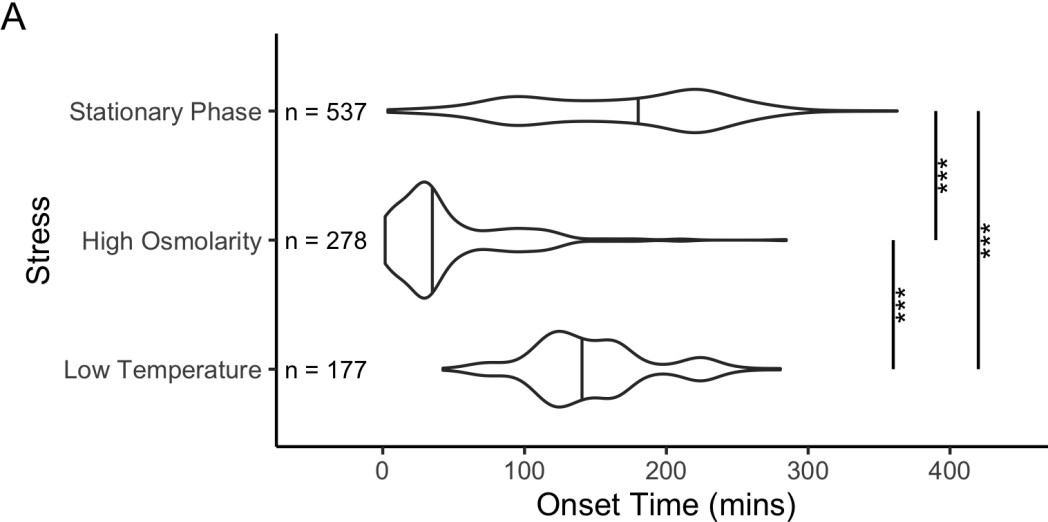

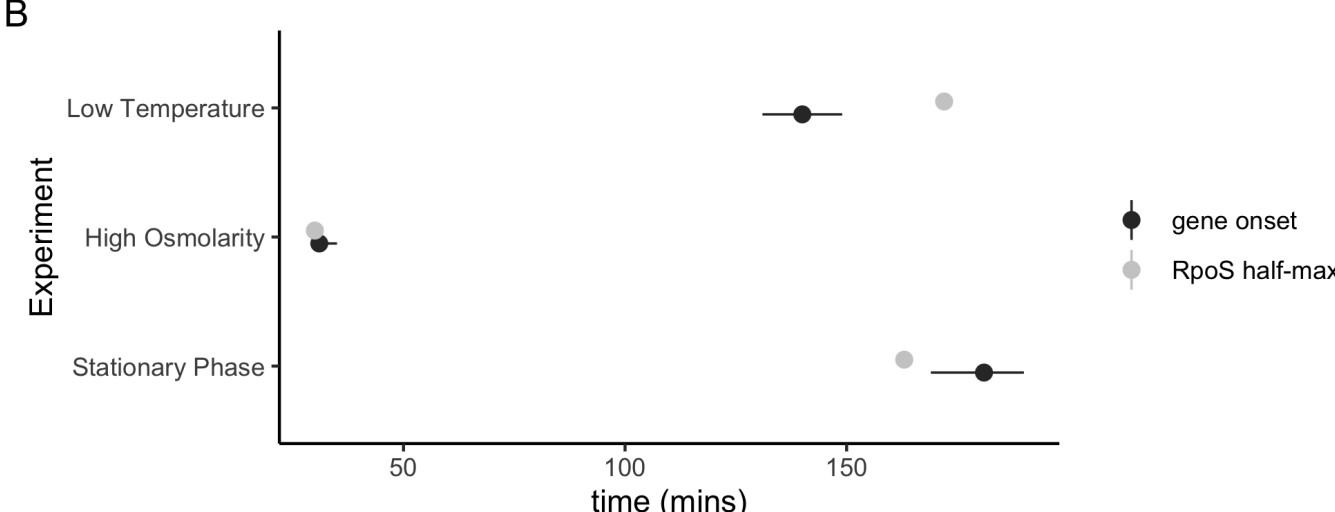

**FIG 4** Onset times of DE genes that were able to be fit by Sicegar. (A) Distribution of onset times of all DE genes in three conditions. Horizontal lines in the middle of the violin plot are the fiftieth percentile of the density estimate. Median onset time was 181 min stationary phase, 31 min for high osmolarity, and 140 min for low temperature. All three median onset times are significantly different (*P* < 0.001, randomization test of difference in medians). (B) Comparison of gene onset times and the time when RpoS reached half maximal levels in that condition. The black circle is the median gene onset time, the black line is the 95% CI on the median (1,000 bootstrap replicates), and the gray dot is the time when RpoS reached half its maximal level.

5). In stationary phase, sensitive genes had a median onset time of 102 min, which was significantly quicker than the value for both linear genes (148 min) and insensitive genes (180 min) (FDR-adjusted *P*-value < 0.01, permutation test for difference in medians, 3,000 permutations). The difference in median onset time between linear and insensitive genes was not significant (FDR-adjusted *P*-value > 0.7).

The sensitivity of a gene (as measured in stationary phase in LB media) can predict the onset time of that gene as it transitions to stationary phase in LB media, so we tested if the same sensitivity classification could be used to predict the timing of expression in response to other stresses, like high osmolarity or low temperature. We found little difference in median onset time of the different classes in high osmolarity or low temperature (Fig. 5). The median onset time in high osmolarity ranged from 29 min in sensitive genes to 36 min for insensitive genes, a nonsignificant difference (FDR-adjusted *P*-value > 0.3, permutation test for difference in medians, 3,000 replicates). The median onset time in low temperature ranged from 139 min for linear genes to

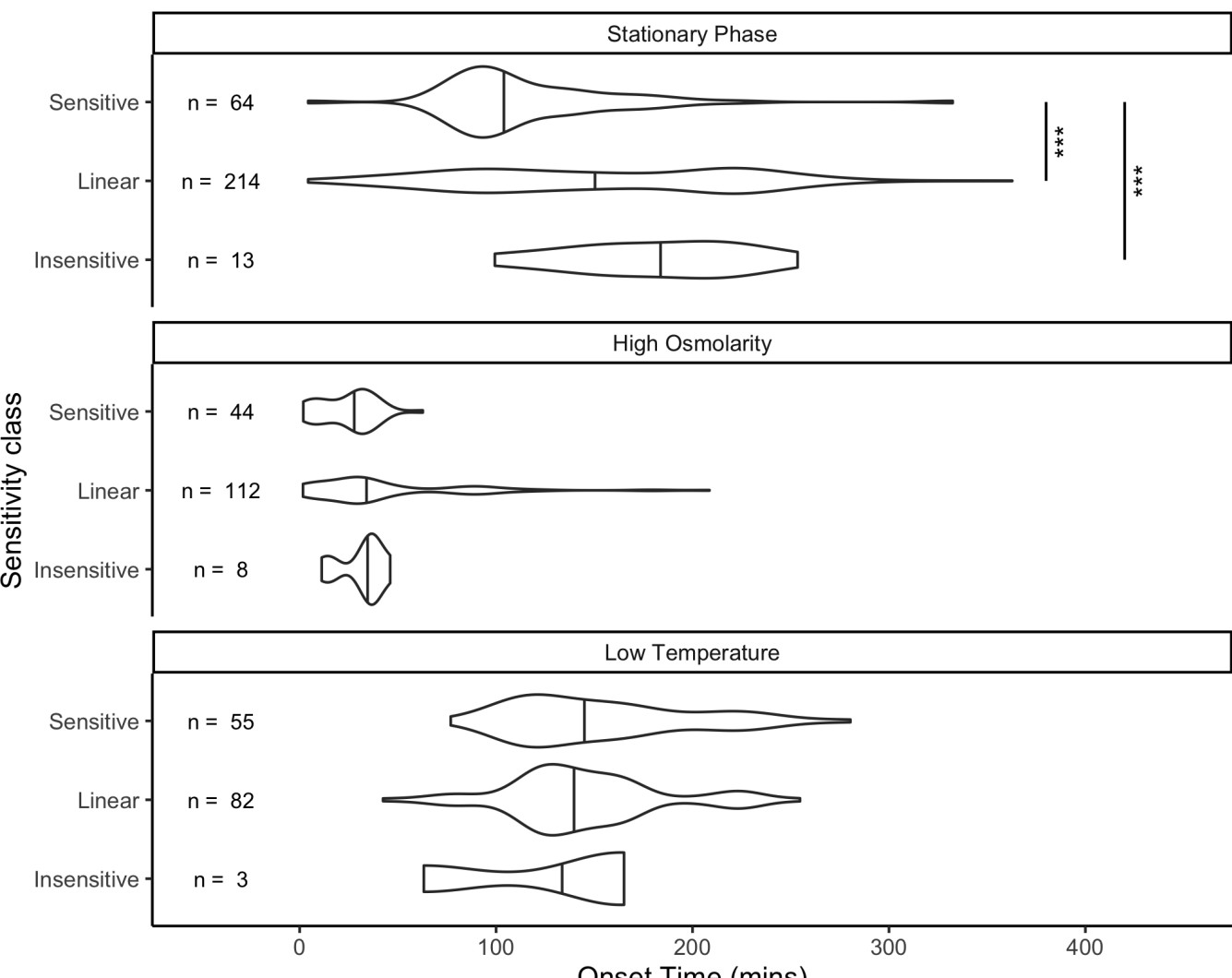

**FIG 5** Distribution of onset times of genes classified as sensitive, insensitive, and linear by reference 15 in each of the three stresses. Vertical lines in the middle of the violin plot are the 50th percentile of the density estimate, which can deviate from the empirical median for small sample sizes. In stationary phase, sensitive genes had a median onset time of 102 min, significantly faster than for linear (148 min) or insensitive (180 min) ($P < 0.001$, randomization test of difference in medians). There was no significant difference in medians between pairs of sensitivity classes for high osmolarity of low temperature.

162 min for insensitive genes, a nonsignificant difference (FDR-adjusted $P$-value > 0.3, permutation test for difference in medians, 3,000 replicates). These results suggest that the classification of sensitivity in stationary phase stress response cannot predict timing of expression of a gene in other stresses.

Since sensitivity measured in stationary phase cannot predict timing in other stresses, it is worth asking if there is any correlation at all between onset times of a gene between two stressors. We examined the onset time for all genes with a successfully fit onset time in each pair of stresses (Fig. 6). There is no observable correlation between onset times in any pair of stresses (absolute value of $r < 0.1$, $P > 0.3$ for all three pairs of stresses). This lack of correlation is consistent with our finding that sensitivity classification in stationary phase can predict timing in that stress but not in others (Fig. 5).

## DISCUSSION

The alternative sigma factor RpoS coordinates the general stress response in *E. coli* by changing patterns of transcription. The timing of RpoS induction differs across stresses.

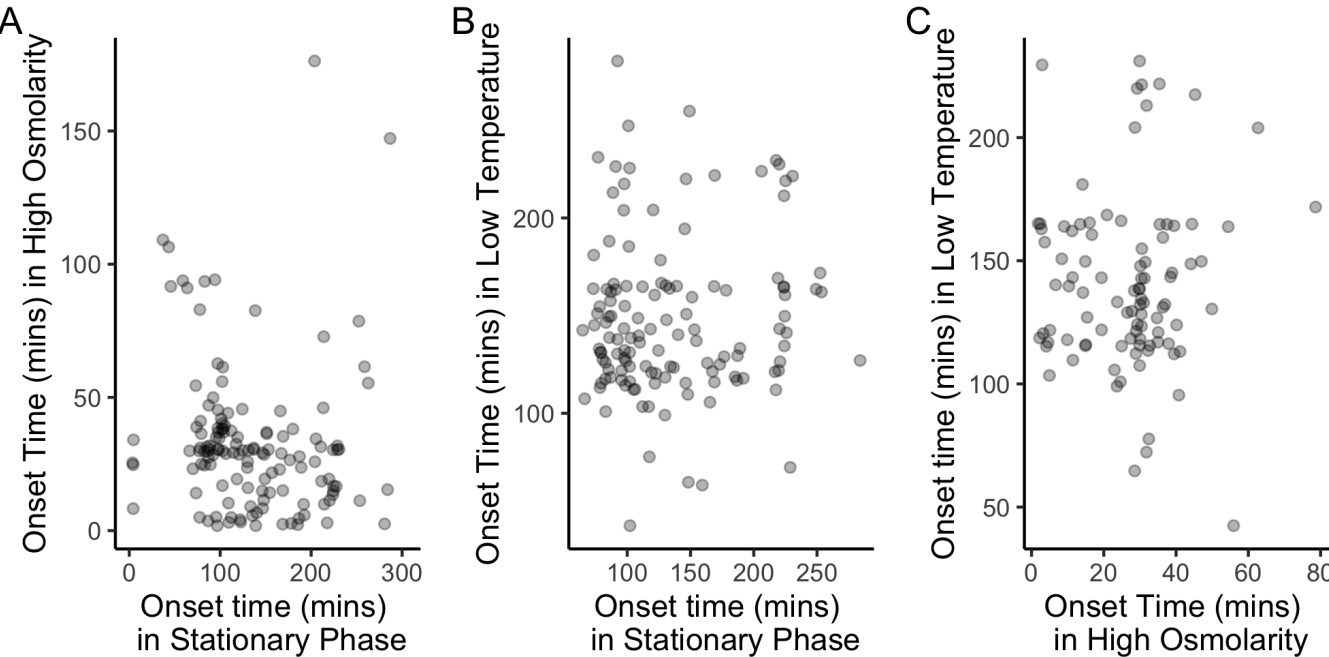

**FIG 6** Onset time is uncorrelated across three stress conditions. (A) Onset times in stationary phase and high osmolarity for $n = 133$ genes with a well-fit onset time fit in both conditions: $r = -0.09$, $P = 0.30$. (B) Onset times in stationary phase and low temperature for $n = 129$ genes with an onset time fit in the two conditions: $r = 0.07$, $P = 0.44$. (C) Onset times in high osmolarity phase and low temperature for $n = 95$ genes with an onset time fit in the two conditions: $r = 0.02$, $P = 0.82$.

The transcriptional response to the change in RpoS levels also differs across stresses, both in terms of which gene's expression changes and the timing of the change. During the entry into stationary phase, we can explain this timing using a simple conceptual model: sensitive genes come on first, insensitive genes come on last. Predictions based on gene sensitivity in stationary phase cannot predict onset timing in other stresses, as onset times in one stress are not correlated with onset times in other stresses.

*E. coli* has previously been reported to accumulate RpoS in response to a wide variety of stresses (1), including the three studied here: entry into stationary phase (32), high osmolarity (35), and low temperature (33). Our work builds on those previous observations by measuring the rates of change of RpoS levels in response to stresses, allowing for quantitative comparisons. A notable finding is that while the peak level of RpoS varied about 2-fold across these stresses, the time it took to reach the maximal level varied nearly 5-fold. Earlier work examining how starvation for different nutrients affected RpoS levels found nearly 10-fold variation between nitrogen starvation and carbon starvation (14). In both nitrogen and carbon starvation, maximal RpoS levels appear to have been reached within 1 h of stress onset, more similar to elevated osmolarity than to low temperature in our study. While molecular mechanisms of RpoS upregulation in response to these stresses have been reported (33, 45–47), converting that molecular understanding to a model of the kinetics of RpoS accumulation remains to be done.

The RpoS regulon in various stresses has also been reported by a number of previous studies (15, 16, 48–50) and our results are broadly consistent with previous findings. For example, we found that there was substantial difference in the genes that are RpoS regulated in different stresses, with only 156 genes DE across all three stresses, similar to the findings of Weber et al. (16) who studied some of the same set of stresses. The work reported here builds on previous studies by measuring the kinetics of the RpoS-dependent transcriptional response to stresses. We found substantial variation in the timing of expression of genes within individual stresses as well as across stresses. In Wong et al. (15), we provided an initial analysis to suggest that differences in sensitivity

of genes resulted in a difference in timing of expression of those genes, but limitations in the underlying data made it impossible to quantify the difference in timing. The results in this study make it clear that there is substantial variation in the onset timing of RpoS-dependent genes during the entry to stationary phase, and that variation can be explained with a simple model: genes that respond strongly to limited levels of RpoS are activated before genes that require higher levels of RpoS to be activated. These results therefore suggest that differences in sensitivity of RpoS-dependent promoters coordinate the transcriptional response of cells entering stationary phase.

Is the variation in onset time among genes physiologically meaningful? In stationary phase, the median difference in onset time between sensitive genes and insensitive genes is 78 min. *E. coli* could be severely harmed in a 78-min time frame if they do not mount a response to a stress. For example, Δ*rpoS E. coli* exposed to high heat for 25 min have nearly 99% reduction in viability, while the viability of wild-type cells is only reduced by about 20% in this time (35). Similarly, Δ*rpoS E. coli* exposed to hydrogen peroxide have viability reduced by about 99% within 10 min, while wild-type cells' viability is reduced by only 20% (35). With stresses affecting viability on this time scale, a difference in onset time of 78 min could be a life-or-death difference for *E. coli* in response to a variety of stresses. The difference in onset timing between sensitive, linear, and insensitive genes is on a physiologically relevant time scale.

The sensitivity of genes to stationary phase RpoS levels does not predict the timing of expression of those gene during other stresses. We found no evidence for a difference in median onset time of the same for sensitive, linear, and insensitive genes in both osmotic shock and low temperature. In addition, onset times are not correlated across the three conditions, implying that genes that come on early in stationary phase are not more or less likely to come on early in high osmolarity or low temperature. These results are easiest to understand if the sensitivity of a gene is not an intrinsic property of its promoter but is instead due to the interactions between the promoter and regulatory proteins that bind it. Consistent with this idea, earlier work found that regions where transcription factors other than RpoS bind to the promoter determine its sensitivity to RpoS (15, 51). Stress-specific responses independent of RpoS almost certainly cause the abundance of other regulatory proteins to differ between the stress conditions in this study. In addition, growth rate is a global regulator of gene expression (52, 53) and we found substantial growth parameter differences between the stresses (Table 2). These differences in global transcription context mean that our classification of promoters as sensitive, insensitive, or linear is most likely condition-dependent, so we should not expect this classification to predict timing across stresses beyond the one in which it was measured. Biologically, this implies that the RpoS-dependent timing of expression can evolve separately across multiple stresses, consistent with the fact that different physiological responses are required for adaptation to different stresses.

In conclusion, we have studied the timing of RpoS-dependent gene expression in *E. coli* across three stresses. We have found that the kinetics of RpoS accumulation in the cell, the identity of genes regulated in a RpoS-dependent manner, and the kinetics of RpoS-dependent transcription vary across the three stresses. During entry into stationary phase, the timing of individual RpoS-dependent genes is associated with how those genes respond to changing levels of RpoS, consistent with a model where the gradual change in RpoS levels in the cell results in different timing of expression of individual genes. However, the timing of expression of individual genes is not correlated across stresses, implying that *E. coli* has distinct timings of RpoS-dependent gene regulation in response to different stresses.

## ACKNOWLEDGMENTS

RNA-Seq libraries were constructed and sequenced at the Broad Institute of MIT and Harvard by the Microbial 'Omics Core and Genomics Platform, respectively. The Microbial 'Omics Core also provided guidance on experimental design and performed

the demultiplexing of RNA-Seq data. Thanks to Lenny Seligman, Danae Schulz, and two anonymous reviewers for feedback and insights.

This material is based upon work supported by the National Science Foundation under Grant No. 1716794.

## AUTHOR AFFILIATIONS

[1]Department of Biology, Harvey Mudd College, Claremont, California, USA
[2]Department of Mathematics and Statistics, Pomona College, Claremont, California, USA

## PRESENT ADDRESS

Johnson Hoang, Doheny Eye Institute, Pasadena, California, USA
Emily Petroni, Division of Molecular and Cellular Biology, Eunice Kennedy Shriver National Institute of Child Health and Human Development, Bethesda, Maryland, USA
Ethan Ashby, Department of Biostatistics, University of Washington, Seattle, Washington, USA

## AUTHOR ORCIDs

Daniel M. Stoebel  http://orcid.org/0000-0002-9461-7567

## FUNDING

| Funder | Grant(s) | Author(s) |
|---|---|---|
| National Science Foundation (NSF) | 1716794 | Daniel M. Stoebel |

## AUTHOR CONTRIBUTIONS

Josephine Adams, Data curation, Formal analysis, Investigation, Writing – original draft, Writing – review and editing | Johnson Hoang, Formal analysis, Investigation, Visualization, Writing – review and editing | Emily Petroni, Formal analysis, Software, Visualization, Writing – review and editing | Ethan Ashby, Formal analysis, Software, Visualization, Writing – review and editing | Johanna Hardin, Formal analysis, Project administration, Writing – review and editing | Daniel M. Stoebel, Conceptualization, Formal analysis, Methodology, Project administration, Supervision, Visualization, Writing – original draft, Writing – review and editing

## DATA AVAILABILITY

The RNA-seq FASTQ files are deposited in the Gene Expression Omnibus with accession number GSE224463. All other data and scripts used for data analysis are at doi:10.5281/ zenodo.8083605.

## ADDITIONAL FILES

The following material is available online.

### Supplemental Material

**Figure S1 (mSystems00663-23-S0001.pdf).** Growth curves of wild-type and ΔrpoS strains during the entry into stationary phase, high osmolarity, and low temperature.
**Figure S2 (mSystems00663-23-S0002.pdf).** Examples of western blots of RpoS levels across the three stresses.
**Legends for supplemental files (mSystems00663-23-S0003.txt).** Legends for supplemental figures and tables.
**Table S1 (mSystems00663-23-S0004.csv).** Values of statistical tests for differential change across time for all genes in wild-type and ΔrpoS strains across entry stationary

phase, high osmolarity, and low temperature and the estimated onset time when it could be estimated.

**Table S2 (mSystems00663-23-S0005.txt).** Biological processes enriched in RpoS-dependent genes in stationary phase.

**Table S3 (mSystems00663-23-S0006.txt).** Biological processes enriched in RpoS-dependent genes in high osmolarity.

**Table S4 (mSystems00663-23-S0007.txt).** Biological processes enriched in RpoS-dependent genes in low temperature.

## Open Peer Review

**PEER REVIEW HISTORY (review-history.pdf).** An accounting of the reviewer comments and feedback.

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
