## [Reviewer comments · mSystems]

The timing of transcription of RpoS-dependent genes varies across multiple stresses in *Escherichia coli* K-12

Josephine Adams, Johnson Hoang, Emily Petroni, Ethan Ashby, Johanna Hardin, and Daniel Stoebel

Corresponding Author(s): Daniel Stoebel, Harvey Mudd College

Review Timeline:

Submission Date:

June 27, 2023

Accepted:

July 14, 2023

Editor: Alejandra Rodríguez-Verdugo

Reviewer(s): The reviewers have opted to remain anonymous.

Transaction Report:

DOI: <https://doi.org/10.1128/msystems.00663-23>

July 14, 2023

Prof. Daniel Stoebel
Harvey Mudd College
Department of Biology, Harvey Mudd College
Claremont, CA 91711

Re: mSystems00663-23 (The timing of transcription of RpoS-dependent genes varies across multiple stresses in *Escherichia coli* K-12)

Dear Prof. Daniel Stoebel:

Thank you for your revisions and for carefully responding to the reviewers' comments. These changes have strengthened the manuscript, which is now ready for publication. Thank you for the privilege of reviewing your work.

Your manuscript has been accepted, and I am forwarding it to the ASM Journals Department for publication. For your reference, ASM Journals' address is given below. Before it can be scheduled for publication, your manuscript will be checked by the mSystems production staff to make sure that all elements meet the technical requirements for publication. They will contact you if anything needs to be revised before copyediting and production can begin. Otherwise, you will be notified when your proofs are ready to be viewed.

If you would like to submit a potential Featured Image, please email a file and a short legend to msystems@asmusa.org. Please note that we can only consider images that (i) the authors created or own and (ii) have not been previously published. By submitting, you agree that the image can be used under the same terms as the published article. File requirements: square dimensions (4" x 4"), 300 dpi resolution, RGB colorspace, TIF file format.

We recognize that the video files can become quite large, and so to avoid quality loss ASM suggests sending the video file via <https://www.wetransfer.com/>. When you have a final version of the video and the still ready to share, please send it to mSystems staff at msystems@asmusa.org.

Sincerely,

Alejandra Rodríguez-Verdugo
Editor, mSystems

Journals Department
E-mail: mSystems@asmusa.org